# Homonymous Hemiatrophy of Macular Ganglion Cell Layer as a Marker of Retrograde Neurodegeneration in Multiple Sclerosis—A Narrative Review

**DOI:** 10.3390/diagnostics14121255

**Published:** 2024-06-14

**Authors:** Larisa Cujbă, Ana Banc, Tudor Drugan, Camelia Alexandra Coadă, Andreea-Petra Cristea, Cristina Stan, Cristina Nicula

**Affiliations:** 1Medical Doctoral School, University of Oradea, 410087 Oradea, Romania; larisacujba@gmail.com; 2Department of Ophthalmology, “Iuliu Hatieganu” University of Medicine and Pharmacy, 400006 Cluj-Napoca, Romania; ana.banc@umfcluj.ro (A.B.); cristrif1959@yahoo.com (C.S.); 3Department of Medical Informatics and Biostatistics, “Iuliu Hațieganu” University of Medicine and Pharmacy, 400349 Cluj-Napoca, Romania; 4Faculty of Medicine, “Iuliu Haţieganu” University of Medicine and Pharmacy, 400012 Cluj-Napoca, Romania; 5Department of Ophthalmology, County Emergency Hospital Cluj-Napoca, 400006 Cluj-Napoca, Romania; andreeapetra30@yahoo.com; 6Department of Maxillo-Facial Surgery and Radiology, “Iuliu Hațieganu” University of Medicine and Pharmacy, 400012 Cluj-Napoca, Romania; niculacristina65@yahoo.com

**Keywords:** homonymous macular hemiatrophy, hemimacular atrophy, macular ganglion cell layer, GCIPL atrophy, optical coherence tomography, multiple sclerosis, retrochiasmal lesions

## Abstract

Retrograde axonal neurodegeneration along the visual pathway—either direct or trans-synaptic—has already been demonstrated in multiple sclerosis (MS), as well as in compressive, vascular, or posttraumatic lesions of the visual pathway. Optical coherence tomography (OCT) can noninvasively track macular and optic nerve changes occurring as a result of this phenomenon. Our paper aimed to review the existing literature regarding hemimacular atrophic changes in the ganglion cell layer identified using OCT examination in MS patients without prior history of optic neuritis. Homonymous hemimacular atrophy has been described in post-chiasmal MS lesions, even in patients with normal visual field results. Temporal and nasal macular OCT evaluation should be performed separately in all MS patients, in addition to an optic nerve OCT evaluation and a visual field exam.

## 1. Introduction

Multiple sclerosis (MS) is the most common inflammatory neurological disease in young adults with the average age of diagnosis around 30 years [1]. Multifocal demyelinating lesions affect both white and grey matter of the central nervous system (CNS) and can lead to irreversible neurological disability [2].

The 2017 McDonald criteria represent the current standard used for diagnosing MS [3]. The diagnosis is based on combined clinical-radiological findings proving the dissemination in time (DIT) and dissemination in space (DIS) of the lesions involving the CNS. DIS refers to detection of characteristic T2-hyperintense lesions observed using magnetic resonance imaging (MRI) in at least two out of four specified regions: periventricular, cortical or juxtacortical, infratentorial, and spinal cord. Additional laboratory examinations such as the testing of oligoclonal bands in cerebrospinal fluid can support MS diagnosis in patients with insufficient clinical or MRI evidence [3].

### 1.1. Visual Pathway Involvement in Multiple Sclerosis

MS demyelinating lesions can affect the afferent visual pathway at all anatomical sites starting from the optic nerve to the primary visual cortex [4]. The anterior visual system originates in the retinal ganglion cells. Their axons collectively compose the optic nerve, optic chiasm, and optic tract (OT), ultimately converging to the lateral geniculate nucleus (LGN) through synaptic connections. Axons of the next-level neurons originating in the LGN form the optic radiations (OR) which project onto the visual cortex; the retrogeniculate visual pathway is also referred to as the posterior visual pathway [5].

Lesions of the visual pathway can initiate the process of neurodegeneration toward the retina (retrograde degeneration) or the occipital cortex (anterograde degeneration). Moreover, trans-synaptic degeneration has been described in the human visual pathway and represents the transmission of neuronal damage effect beyond the LGN—i.e., the synaptic relay that connects the anterior and posterior visual pathways [5,6].

Optic neuritis is one of the most common presenting syndromes in MS [7]. In 2016, it was proposed that the optic nerve should be considered the fifth anatomical site of MS lesions in order to fulfil MRI criteria for DIS [8]. However, MRI, visual evoked potentials, and optical coherence tomography (OCT) are considered insufficient to support the diagnosis of MS in patients without a clear history or a clinical proof of optic neuritis [3].

Although rare, optic chiasmitis can also occur in MS, leading to bilateral visual loss that is either simultaneous or rapidly sequential, and a normal fundus in the initial stage, while the MRI examination usually demonstrates optic chiasm enhancement [9].

The damage of the OT in MS is poorly understood since these structures are not part of the classically evaluated sites for lesion dissemination [10,11]. Postmortem examinations presented by Plant et al. in 1992 revealed the lack of direct involvement of the OT in such patients [12]. However, subsequent autopsy studies investigating the presence of axonal damage at this level demonstrated a roughly 30% decrease in axonal density in MS patients, without obvious alteration of the cross-sectional appearance of the optic nerves [13]. Dasenbrock et al. emphasized that OT damage can appear in the absence of direct lesions, similarly to the optic nerve damage in the absence of optic neuritis [11]. Although direct OT lesions are rare in MS, studies reveal secondary damage at this level caused by either optic neuritis via anterograde degeneration, or higher lesions of the posterior visual pathway via retrograde trans-synaptic degeneration [11,14].

LGN volume loss in MS also occurs due to anterograde degeneration initiated from the retina, or retrograde degeneration from lesions involving the optic radiations [15].

Not only can retrogeniculate visual pathway damage occur, but the posterior visual pathway is also a preferentially affected area in MS [16]. Postmortem histopathological analyses along with MRI studies revealed that lesions of the OR are a common finding in MS patients [17,18]. The involvement of the OR is explained by their localization through the periventricular white matter, at which site MS lesions typically occur [11]. However, the secondary neurodegeneration of OR stemming from distant visual system impairment such as optic neuritis has also been described [5]. OR have a paramount role in the transmission of visual information to the primary visual cortex. Any damage at this level contributes in a great manner to the visual disability in MS [14,19,20].

Atrophy of the visual cortex has been described in individuals with MS with a history of optic neuritis, but typical MS lesions within the primary visual cortex have also been reported [5,16,21,22].

### 1.2. The Role of OCT in Tracking Neurodegeneration

Ocular OCT is a fast high-resolution imaging technique that can identify lesions localized at any level of the visual pathway by assessing the retinal neuronal damage at axonal (retinal nerve fiber layer) and/or cellular body (ganglion cell layer) level [23].

MS is associated with peripapillary retinal nerve fiber layer (pRNFL) thinning and macular ganglion cell (mGC) loss, with changes being greater in eyes with prior optic neuritis than in those with no history of such. Therefore, both OCT markers are recommended for the diagnosis and monitoring of MS, as well as for research purposes [24]. Of note, some papers report the OCT macular thickness of ganglion cell layer only (mGCL), while others present the combined thickness of ganglion cell layer plus inner plexiform layer (mGC-IPL), depending on the OCT machine employed.

The mechanism of retinal axonal atrophy and mGC loss in MS in the absence of optic neuritis remains unknown. Possible causative mechanisms are as follows: diffuse neurodegenerative processes, the presence of subclinical inflammation or chronic demyelination along the visual pathway, or retrograde trans-synaptic degeneration following lesions of the OR [5,25].

A retrochiasmal lesion can determine the atrophy of retinal ganglion cells that results in a homonymous hemimacular mGC loss on OCT, and corresponding homonymous visual field (VF) defects. The mGC changes appear earlier in the case of an OT lesion than in a retrogeniculate injury [26]. Mühlemann et al. reported that the homonymous hemiatrophy of mGC becomes discernible approximately 5 months following the onset of a retrochiasmal lesion [26]. The degree of mGC reduction in OR lesions varies notably among individuals and might reach its highest point within the initial two years post-injury [27].

In their study on retrogeniculate lesions linked to homonymous hemianopia, Jindahra et al. observed that pRNFL thinning becomes apparent within the first few months [28]. This thinning worsens rapidly during the first 1–2 years, after which it tends to stabilize or progress slowly in the following years [28]. Additionally, the authors observed that visual recovery is likely to appear in the patients who experienced less pRNFL thinning in the first years [28]. The size of the lesion, but most importantly, its strategic localization (e.g., LGN), can have a paramount influence on the magnitude of trans-synaptic neurodegeneration [28].

Post-chiasmal retrograde degeneration presents some distinct features in clinical practice, such as the following: congruous mGC changes respecting the vertical midline, discrepancy between VF/mGC changes and pRNFL values, or the asymmetry of pRNFL (more evident in the superior-nasal and/or inferior-nasal sectors) [29]. Several studies showed that mGC evaluation is of greater importance in detecting retrograde trans-synaptic degeneration compared to pRNFL [26]. In contrast, pre-chiasmal neurodegeneration presents with incongruous changes in the mGC assessment that tend to respect the horizontal midline, lack of concordance between VF/mGC analysis and pRNFL, and a symmetrical pRNFL thinning pattern [29].

In the following sections of this paper, we will refer to post-chiasmal MS lesions.

## 2. Macular OCT Changes in MS

### 2.1. Homonymous Macular Hemiatrophy in MS

Several studies reported atrophic mGC changes respecting the vertical midline in patients with MS without a history of optic neuritis—Table 1 [30,31,32,33,34]. The majority of patients presented lesions involving the OR, with variable VF findings (Table 1).

Huang-Link et al. [30] reported in their article two cases of MS patients both showing a demyelinating lesion in the left OR related to a new clinical relapse. The OCT report of the first case showed left homonymous hemimacular mGC-IPL thinning along with pRNFL thinning, as well as a right homonymous hemianopia on VF test, corresponding to the left retrochiasmal lesion on brain MRI. The second case showed a progressive development of a nasal hemimacular mGCL atrophy and a slight thinning of pRNFL in the right eye along with normal VF, while the left eye showed generalized mGC loss corresponding to an older episode of optic neuritis.

Al-Louzi et al. [31] reported six RRMS cases with no prior optic neuritis, showing MRI lesions in different areas of the posterior visual pathway (such as LGN, OR, or visual cortex). This led to the development of homonymous mGC-IPL atrophy on OCT scans, with or without typical changes in homonymous hemianopia.

In their article, Lukewich et al. [32] presented seven patients, including four with demyelinating conditions (three of whom had MS) and three with traumatic brain injuries (TBI). Macular OCT examination revealed the homonymous hemiatrophy of mGC-IPL without significant VF impairment. Among patients with demyelinating conditions, lesions were observed at the OT level in three cases and at the LGN level in one case. Additionally, three of these patients experienced homonymous hemianopia, which resolved over time. In patients with TBI, no visible retrochiasmal lesions were observed on brain MRI.

In the study conducted by Ilardi et al. [33], 5 out of 135 patients with demyelinating diseases (MS, neuromyelitis optica spectrum disorder, and myelin oligodendrocyte glycoprotein antibody disease) showed homonymous hemimacular mGC-IPL atrophy, representing only 3.7% of the sample. Another pattern of mGC-IPL thinning involving a combination of hemimacular thinning in one eye and diffuse macular thinning in the other eye (“whole + half pattern”) was observed in 8.1% of the patients [33]. This pattern, particularly involving the nasal retina in the eye with the hemimacular thinning, may result from the chiasmal disease affecting all fibers of the optic nerve in one eye and the crossing fibers originating from the other eye. Among these 11 patients, 2 had a documented history of optic neuritis, while for the remaining patients, the authors could not determine whether the OCT findings were due to chiasmal involvement or optic nerve disease [33]. Most patients diagnosed with homonymous hemimacular atrophy exhibited demyelinating lesions in the post-geniculate visual pathway spread throughout both brain hemispheres, with the exception of one individual [33]. The “whole + half” aspect on macular OCT may likely be indicative of undetected or subclinical optic neuritis in one eye, compounded by a retrochiasmal lesion. Given the authors’ acknowledgment that their study was constrained by the inability to thoroughly evaluate the optic nerves, chiasm, and tracts on MRI, the accuracy of the reported percentage remains uncertain. Considering optic neuritis is a common manifestation of MS; it may have influenced the results, as many patients may not recognize mild episodes. Therefore, acknowledging the characteristic pattern of “homonymous hemiatrophy” associated with post-chiasmal lesions is crucial. The presence of homonymous hemiatrophy may be partially masked by a global loss of ganglion cells due to optic neuritis, potentially leading to an underestimation of its true prevalence, particularly given the rarity of chiasmal lesions in MS compared to the frequent occurrence of optic neuritis. In Figure 1, we present two MS patients with different macular GCL thinning patterns.

Schmutz and Borruat [34] conducted a retrospective study on MS patients with VF defects using automated static perimetry. Ten patients (50%) also underwent macular assessment, with OCT revealing homonymous mGCL thinning in three patients, diffuse unilateral or bilateral thinning of mGCL following previous optic neuritis in six patients, and normal mGCL thickness in one patient. Out of 20 patients included in the study, most had RRMS (90%). Homonymous VF defects (i.e, homonymous hemianopia, homonymous quadrantanopia, or homonymous scotoma) were the presenting symptoms in 35% of patients [34]. Sixty percent experienced complete recovery within a median time of 10 weeks. MRI showed lesions (located at the level of OR, OT, or LGN) explaining the VF defects in 7 out of 11 patients [34]. The authors also noted that homonymous hemiatrophy could occur due to either direct retrograde axonal degeneration or trans-synaptic retrograde axonal degeneration [34].

### 2.2. Relationship between OCT and VF

Retrochiasmal lesions most often lead to nonpersistent or even asymptomatic homonymous VF defects. It has been noted that in patients with MS, only large post-chiasmal lesions can lead to symptomatic homonymous VF defects [12].

Studies reveal that among patients presenting with homonymous VF defects, the hemiatrophy of mGC is found in 95% of those presenting pre-geniculate lesions, in comparison to only 64% of patients with retrogeniculate lesions [26].

Although rare, homonymous hemiatrophy of the mGCL in the absence of significant VF defects can be a hallmark of previous retrochiasmal lesions in MS patients, especially in those without obvious corresponding lesions on the MRI scan or a history of TBI [32].

The OCT evaluation of mGC can be a useful tool for the evaluation of DIT and DIS of MS characteristic lesions, as the homonymous hemiatrophy of mGC appears in more than 70% of all retrochiasmal damage [26,32]. Although homonymous hemiatrophy of mGC is more sensitive in detecting retrograde trans-synaptic degeneration than pRNFL assessment alone, and there are reports of mGC thinning associated with normal pRNFL values, it is recommended that both indicators be analyzed concurrently for the detection of retrochiasmal lesions [26]. The lack of specificity of pRNFL to detect sectoral neural loss seems to be attributed to the intricate arrangement of the nerve fibers around the optic disc [35].

## 3. Discussion

### 3.1. OCT Findings in MS

Kupersmith et al. [36] demonstrated that mGC-IPL is better than pRNFL in detecting early atrophy due to optic neuritis. The average pRNFL atrophy related to optic neuritis measured approximately 20 μm. Moreover, swelling of the optic disc may interfere and alter precise pRNFL assessments. mGC-IPL is a reliable biomarker for the detection of neurodegeneration in the visual pathway and can depict atrophy sooner than pRNFL. The mGC-IPL thinning can be detectable within 1 month post an acute episode of optic neuritis, whereas the detection of pRNFL changes may take up to 3 months [24].

Gabilondo et al. [37] studied retinal changes secondary to optic neuritis. They observed that optic neuritis-related macular atrophy was more severe and uniform at the level of internal sectors (i.e., surrounding the fovea) of the ETDRS grid, reflecting the mGC-IPL thinning. External sectors corresponding to the papillo-macular bundle presented more marked atrophy, mainly due to macular RNFL (mRNFL) thinning. Petzold at al. [38] did not recommend the use of mRNFL as a monitoring marker due to possible automated segmentation errors related to common ocular disorders (e.g., epiretinal membrane).

Irreversible damage to axons along the visual pathway can lead to retrograde trans-synaptic axonal degeneration, causing atrophy of the inner retinal layers [24]. This degeneration stops at the inner nuclear layer (INL)—the first bipolar neuron of the visual pathway—which acts as a barrier, thus making the INL of interest for the investigation of inflammation [24]. Gabilondo et al. [37] observed that INL thickness seems to thicken after an episode of optic neuritis. Balk et al. [39] also showed a link between increased INL volume, acute optic neuritis, and clinical relapses in MS eyes. Saidha et al. [40] reported the association of MS activity and a greater thickness of the combined INL and outer plexiform layer, while Knier et al. [41] observed a sustained INL thickness reduction after the use of disease modifying treatment. INL thickening can also be associated with the existence of microcystic macular oedema [42]. On the contrary, Green et al. [43] reported evident INL atrophy in non-acute progressive or longstanding MS eyes.

OCT-angiography (OCT-A) has been used to investigate the retinal vessel density and blood flow alterations in MS patients [44]. The observed retinal vessel density reduction in MS eyes may be due to inflammation of the optic nerve and retina, or neuroaxonal degeneration and thinning of retinal layers thickness [44]. A systematic review and meta-analysis of studies on MS patients using OCT-A concluded that MS eyes present a reduced vessel density in the superficial capillary plexus (SCP), deep capillary plexus, and radial peripapillary capillary plexus [44]. Notably, Murphy et al. [45] have shown a correlation between macular SCP vessel density and mGC-IPL thickness, particularly in MS-related optic neuritis eyes, indicating the importance of the SCP in supplying blood to the RNFL and mGC-IPL, as reported by Park et al. [46]. However, this meta-analysis did not show significant differences in macular SCP vessel density between MS eyes with and without optic neuritis [44]; therefore, OCT-A alone cannot be used as a marker to differentiate between optic neuritis and non-optic neuritis eyes.

### 3.2. OCT Parameters as an Outcome Measure in MS Clinical Trials

Although recent clinical trials included the OCT evaluation in the assessment of patients with different types of MS, to the best of our knowledge, there is no trial that investigated the hemimacular OCT changes respecting the vertical midline. In the STRIVE study [47], the only included OCT parameter was pRNFL. In the SPRINT-MS study [48], the investigated macular OCT parameters were the total macular volume and the mean mGC-IPL thickness. In the post hoc analysis of the same trial, the researchers investigated the average mGC-IPL thickness, as well as the average INL and outer nuclear layer thicknesses [49]. Another study included the mean values of pRNFL, mRNFL, and mGC-IPL thicknesses [50]. Moreover, the OCT results of only one eye per patient were considered [50].

By only including the average values of the retinal layers’ thicknesses in the analysis, a sectoral thinning such as hemimacular atrophy can be missed. We suggest that nasal and temporal macular sectors should be assessed separately as the retrograde neurodegeneration process initiated by retrochiasmal MS lesions can determine an asymmetrical macular alteration relative to the vertical midline. This idea is also supported by the results of the study conducted by Pérez del Palomar et al. [51], in which pRNFL, mRNFL, and mGC-IPL losses were compared between MS patients and healthy controls using different machine learning techniques. According to this research, the temporal area of the macula was highly significant to predict the disease [51].

### 3.3. Differences in OCT Protocols for the Evaluation of Retrochiasmal Lesions in MS

Although the combined analysis of pRNFL and mGC-IPL is recommended for the detection and monitoring of neurodegeneration secondary to post-chiasmal visual pathway lesions in MS [24], some papers have not provided complete data on both OCT markers. Moreover, some studies have used mGCL analysis alone, whereas others evaluated the GCL thickness in combination with the IPL layer [30,31,32,33,34].

For the macular analysis of patients with MS, the most frequently used scanning protocol has been the ETDRS grid with concentric circles [30,31,32,33,34]. Recently, there has been a transition to using a square grid for these examinations [52,53].

### 3.4. Homonymous Hemiatrophy of mGC in Other Disorders

The homonymous hemiatrophy of mGC is not specific to demyelinating etiology, but rather serves as a distinctive hallmark indicating the presence of retrograde degeneration in a spectrum of cerebral disorders such as those of vascular, tumoral, compressive, traumatic, inflammatory, or demyelinating origins.

Meier et al. [54] reported the presence of mGC thinning in retrogeniculate vascular, inflammatory, and compressive lesions. In their study, all patients had a homonymous hemiatrophy pattern of mGC loss that correlated well with the VF loss, but not with the pRNFL thickness [52]. However, the prediction of the extent of VF loss from mGC loss and vice versa was not readily achievable. The temporal characteristics for trans-synaptic retrograde degeneration remained indeterminate, as its occurrence was not consistently manifested in all individuals affected by retrogeniculate lesions [54].

Blanch et al. emphasized the importance of OCT macular assessment in patients with sellar tumors [55]. Despite normal or near normal VF results or pRNFL thickness, the mGC-IPL evaluation showed binasal loss or global mGC-IPL thinning, which demonstrates that mGC-IPL has the finest sensitivity in the detection of chiasmal compression [55]. The mGC-IPL changes can be detected as early as 3 to 4 weeks after the axonal injury, whereas pRNFL thinning can be identified 4 to 6 weeks post-injury [56].

The utility of mGC analysis in detecting retrograde degeneration is also supported by several other papers [26,57,58,59].

## 4. Future Directions

We suggest that the difference between nasal and temporal macular thicknesses (irrespective of the evaluated macular layer) should be investigated in MS patients compared to healthy controls and/or between various subtypes of MS. Moreover, both eyes of the patients should be included in the analysis. An asymmetry score between right-sided and left-sided hemimaculae could prove to be of benefit for the early detection of retrochiasmal MS lesions using OCT technology.

## 5. Conclusions

MS lesions can affect the entire afferent visual pathway, and a comprehensive ophthalmic evaluation of MS patients is mandatory even in the absence of symptoms of optic neuritis. The OCT evaluation of mGC thickness in addition to pRNFL thickness and VF evaluation is of paramount importance. Based on the available data in the literature, we suggest that not only should the average mGC thickness be assessed in MS patients, but also the nasal and temporal mGC thicknesses should be assessed, as hemimacular mGC atrophy can be the first sign of a retrochiasmal MS lesion.

## Figures and Tables

**Figure 1 diagnostics-14-01255-f001:**
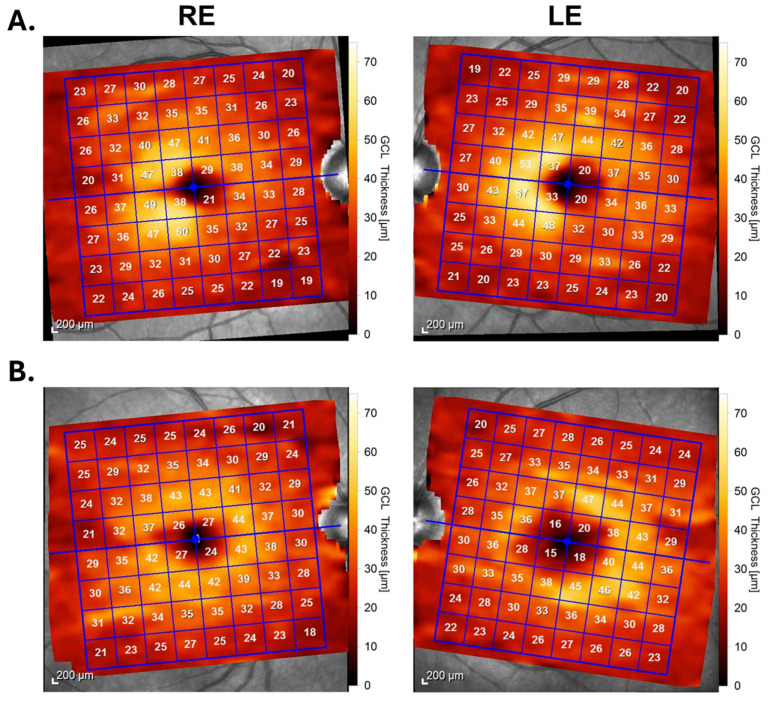
Spectralis SD-OCT (spectral domain optical coherence tomography) macular scans (Heidelberg Engineering, Heidelberg, Germany) using Posterior Pole Analysis. (**A**) Macular analysis shows nasal hemimacular GCL thinning in the RE and temporal hemimacular GCL thinning in LE (left homonymous hemiatrophy of the GCL). (**B**) Macular analysis shows combination of diffuse macular GCL thinning in RE and nasal hemimacular GCL thinning in the LE (“whole + half pattern”). GCL, ganglion cell layer; RE, right eye; LE, left eye.

**Table 1 diagnostics-14-01255-t001:** Summary of the studies reporting mGC hemiatrophy in MS.

Authors, Year of Publication	Number of Patients	Prior Episodes of Optic Neuritis (Number of Eyes)	VF Findings	mGC Thinning Pattern	pRNFL Findings	MRI Findings
Huang-Link et al., 2014 [30]	Active MS: 2	0	Right homonymous hemianopia	Left homonymous mGC atrophy	Bilateral pRNFL thinning (most marked temporally)	Demyelinating lesion in the left OR
		1	Normal	Right eye: nasal mGC thinning Left eye: diffusely thinned mGC	Slightly thinned pRNFL in the right eye and markedly thinned pRNFL in the left eye (most marked temporally)	Demyelinating lesion in the left OR
Al-Louzi et al., 2017 [31]	RRMS: 6	0	4 pts: homonymous hemianopia	1 pt: left homonymous mGC loss 3 pts: right homonymous mGC reduction 2 pts: right homonymous mGC reduction	Not reported	1 pt: left occipital white matter lesion 3 pts: right occipital white matter lesion 2 pts: right thalamic lesions
Lukewich et al., 2020 [32]	Demyelinating disease: 4 (MS: 3) TBI: 3	Not reported	No significant defect at the time of evaluation	4 pts: right homonymous mGC thinning 3 pts: left homonymous mGC thinning	Not reported	3 pts: contralateral OT lesion 1 pt: LGN lesion
Ilardi et al., 2020 [33]	RRMS: 4 NMO: 1	1	1 pt: history of homonymous hemianopia	2 pts: right homonymous mGC thinning 3 pts: lef homonymous mGC thinning	Not reported	3 pts: bilateral demyelinating lesions of the post-geniculate visual pathway 1 pt: lesion of the OR on the opposite expected site 1 pt: no visible demyelinating lesion
Schmutz and Borruat, 2020 [34]	RRMS: 18 PPMS: 1 SPMS: 1	9	3 pts: homonymous VF defect 11 pts: homonymous quadrantanopia 2 pts: homonymous hemianopia	3 pts: homonymous thinning 6 pts: diffuse unilateral or bilateral thinning (resulting from previous episodes of optic neuritis) 1 pt: normal mGC thickness	Not reported	7 pts: an MRI-defined lesion explaining the VF defects: 5 pts—OR lesion; 1 pt—OT lesion; 1 pt—LGN lesion

RRMS, relapsing–remitting MS; TBI, traumatic brain injury; NMO, neuromyelitis optica; PPMS, primary progressive MS; SPMS, secondary progressive MS; mGC, macular ganglion cell layer; MS, multiple sclerosis; pRNFL, peripapillary retinal nerve fiber layer; pt(s), patient(s); VF, visual field; OT, optic tract; LGN, lateral geniculate nucleus; OR, optic radiations.

## Data Availability

Not applicable.

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
