# Peer review of "Homonymous Hemiatrophy of Macular Ganglion Cell Layer as a Marker of Retrograde Neurodegeneration in Multiple Sclerosis—A Narrative Review"

_diagnostics, 2024, doi:10.3390/diagnostics14121255_

Round 1

Reviewer 1 Report

Comments and Suggestions for Authors

The purpose of this article was to summarize the literature related to homonymous hemi-macular atrophy in multiple sclerosis (MS) patients without previously diagnosed optic neuritis. The authors emphasize the need for more comprehensive ophthalmic evaluation in MS patients using optical coherence tomography (OCT). OCT evaluation of the thickness of macular ganglion cell layer is indicated as an important marker of retrograde neurodegeneration in MS patients. Detection of this marker improves MS diagnosis and disease monitoring.

In general, the article reviews the topic in detail, as well as includes the latest scientific research on macular ganglion cell layer hemiatrophy in MS. It should be mentioned that the readability of the text is somewhat difficult due to the writing style - the structure of individual sentences is overly complicated. Minor grammatical errors appear in the text, but they do not affect the scientific quality of the article and can be easily improved.

This article structurally resembles the descriptive part of the monograph, as it lacks parts - methodology and results. In the context of the literary review, there should be a methodical part, where the methodology and criteria for the selection of literary sources are described, and in the result part, the analysis of the selected literary sources is carried out with a specific goal, set in the introduction.

Title and Abstract

The title of the article reflects the content of the article well, but the combination of words "hemi-atrophy" would be advised to be written using the combined form without the hyphen as "hemiatrophy", which meets the standard of medical terminology. The abstract gives an adequate brief overview of the content of the review article.

Introduction

The introduction is incomplete and should be improved.

It mainly describes the disease (MS) and its diagnostic aspects, but it lacks a brief and comprehensive summary of the topic of the article and a rationale for addressing this topic, focusing on why it is important and why a review is necessary. The reader lacks information about what will be discussed in the article and what the main focus will be.

Line 36

“young adults [1]” -  it is necessary to specify the age.

There is a grammatical error in the word "imagining" which should be written as "imaging" (line 42).

SPECIFIC POINTS TO BE ADDRESSED:

-         The term “optic radiations” is mentioned for the first time in line 53 and has no abbreviation, but it is mentioned again in lines 81 and 84 and only in the line 84 is it given the abbreviation "OR" and also it is not clear why the term is in italics in this line. The corresponding abbreviation should be given when the term is first mentioned in the text to make it easier for the reader to the reader to follow the text.

-       Terms in italics should be revised for consistency. For example, the term "optic neuritis" in line 60 is italicized, while it is not italicized in lines 88, 176, 189, and 191. the same applies to the term "visual cortex" in lines 53 and 91 - 92, as well as "optic tracts" in line 69. It should also be considered whether italics are even unnecessary in some cases. For example, in line 80, the abbreviation "LGN" is in italics, which is confusing.

-       The sentence lines 82 - 83 should be restructured to improve its comprehensibility.

-       The sentence in lines 119 - 122 could benefit from a slight adjustment for clarity and readability. "In their study investigating retrogeniculate lesions associated with homonymous hemianopia, Jindahra et al. noticed that the pRNFL thinning becomes evident within the first few months, with a high rate of worsening in the first 1-2 years, and then tends to stabilize or evolve slowly in the subsequent years”.

-       lines 182- 183: “Given the authors’ acknowledgment that their study was constrained by the inability to thoroughly evaluate the optic nerves, chiasm, and tracts on MRI, the accuracy of the reported percentage remains uncertain.”  - This statement is the subject of discussion and is not part of the review results.

 The summary of the studies, which is reflected in Table 1, gives a well-reviewed summary of the research results. It can be recommended to put square brackets with the reference number after the author or publication in Table 1 so that it would be easier to find this publication in the list of references. Since the table already contains a lot of text, it should be checked that it uses all possible abbreviations consistently. For example, the usage of the full disease name "multiple sclerosis" is not necessary if it already has an abbreviation, which is clarified below the table.

Discussion

The discussion chapter is divided into four subsections and discusses important aspects of OCT in MS clinical studies and the parameters studied. Based on the results of these studies, the authors provide valuable suggestions on how to improve the analysis of OCT results, considering the possibility of hemi-macular atrophy. However, considering that it is not indicated anywhere which publications are directly included in the introduction part, it is also unclear whether the information in the discussion is the subject of discussion or a continuation of the introduction.

A small correction in the line 245 would be necessary – after the abbreviation "(INL)" there are two dashes in a row.

Conclusions and future directions

In these chapters, the authors indicate the identified gaps in existing knowledge, offering insights on future research directions showing that the review not only synthesizes existing knowledge but also contributes to advancing the field.

Considering all the above comments, this publication can be recommended for publication after a major revision.

Author Response

The purpose of this article was to summarize the literature related to homonymous hemi-macular atrophy in multiple sclerosis (MS) patients without previously diagnosed optic neuritis. The authors emphasize the need for more comprehensive ophthalmic evaluation in MS patients using optical coherence tomography (OCT). OCT evaluation of the thickness of macular ganglion cell layer is indicated as an important marker of retrograde neurodegeneration in MS patients. Detection of this marker improves MS diagnosis and disease monitoring.

In general, the article reviews the topic in detail, as well as includes the latest scientific research on macular ganglion cell layer hemiatrophy in MS. It should be mentioned that the readability of the text is somewhat difficult due to the writing style - the structure of individual sentences is overly complicated. Minor grammatical errors appear in the text, but they do not affect the scientific quality of the article and can be easily improved.

Thank you for your observation. We have reviewed our manuscript for grammatical errors.

This article structurally resembles the descriptive part of the monograph, as it lacks parts - methodology and results. In the context of the literary review, there should be a methodical part, where the methodology and criteria for the selection of literary sources are described, and in the result part, the analysis of the selected literary sources is carried out with a specific goal, set in the introduction.

Thank you for your suggestion. However, this paper is a narrative review, not a systematic review. The sections you suggested are part of a systematic review, and we did not intend to conduct this type of review.

Title and Abstract

The title of the article reflects the content of the article well, but the combination of words "hemi-atrophy" would be advised to be written using the combined form without the hyphen as "hemiatrophy", which meets the standard of medical terminology. The abstract gives an adequate brief overview of the content of the review article.

Thank you for your observation. We reviewed the terminology and made the necessary changes.

Introduction

The introduction is incomplete and should be improved. It mainly describes the disease (MS) and its diagnostic aspects, but it lacks a brief and comprehensive summary of the topic of the article and a rationale for addressing this topic, focusing on why it is important and why a review is necessary. The reader lacks information about what will be discussed in the article and what the main focus will be.

Thank you for your suggestion. We have expanded the Introduction section by reorganizing the following sections.

Line 36 “young adults [1]” - it is necessary to specify the age.

Thank you for your suggestion. We have included the information.

There is a grammatical error in the word "imagining" which should be written as "imaging" (line 42).

Thank you for your observation, we have corrected the spelling error.

SPECIFIC POINTS TO BE ADDRESSED:

The term “optic radiations” is mentioned for the first time in line 53 and has no abbreviation, but it is mentioned again in lines 81 and 84 and only in the line 84 is it given the abbreviation "OR" and also it is not clear why the term is in italics in this line. The corresponding abbreviation should be given when the term is first mentioned in the text to make it easier for the reader to the reader to follow the text.

Thank you for your observation. We have made the necessary changes.

Terms in italics should be revised for consistency. For example, the term "optic neuritisin line 60 is italicized, while it is not italicized in lines 88, 176, 189, and 191. the same applies to the term "visual cortex" in lines 53 and 91 - 92, as well as "optic tracts" in line 69. It should also be considered whether italics are even unnecessary in some cases. For example, in line 80, the abbreviation "LGN" is in italics, which is confusing.

Thank you for your observation. To avoid confusion, we have ensured that no words appear in italics.

The sentence lines 82 - 83 should be restructured to improve its comprehensibility.

Thank you for your comment. We have rephrased the sentence.

The sentence in lines 119 - 122 could benefit from a slight adjustment for clarity and readability. "In their study investigating retrogeniculate lesions associated with homonymous hemianopia, Jindahra et al. noticed that the pRNFL thinning becomes evident within the first few months, with a high rate of worsening in the first 1-2 years, and then tends to stabilize or evolve slowly in the subsequent years”.

Thank you for your observation, we have rephrased the sentence.

Lines 182- 183: “Given the authors’ acknowledgment that their study was constrained by the inability to thoroughly evaluate the optic nerves, chiasm, and tracts on MRI, the accuracy of the reported percentage remains uncertain.”  - This statement is the subject of discussion and is not part of the review results.

We thank you for your comment, but we are not following the structure of a systematic review article.

The summary of the studies, which is reflected in Table 1, gives a well-reviewed summary of the research results. It can be recommended to put square brackets with the reference number after the author or publication in Table 1 so that it would be easier to find this publication in the list of references. Since the table already contains a lot of text, it should be checked that it uses all possible abbreviations consistently. For example, the usage of the full disease name "multiple sclerosis" is not necessary if it already has an abbreviation, which is clarified below the table.

Thank you for your suggestions. We have made the necessary changes.

Discussion

The discussion chapter is divided into four subsections and discusses important aspects of OCT in MS clinical studies and the parameters studied. Based on the results of these studies, the authors provide valuable suggestions on how to improve the analysis of OCT results, considering the possibility of hemi-macular atrophy. However, considering that it is not indicated anywhere which publications are directly included in the introduction part, it is also unclear whether the information in the discussion is the subject of discussion or a continuation of the introduction.

Thank you for your comment.

A small correction in the line 245 would be necessary – after the abbreviation "(INL)" there are two dashes in a row.

Thank you for your observation, we have corrected the spelling mistake.

Conclusions and future directions

In these chapters, the authors indicate the identified gaps in existing knowledge, offering insights on future research directions showing that the review not only synthesizes existing knowledge but also contributes to advancing the field.

Considering all the above comments, this publication can be recommended for publication after a major revision.

We kindly thank you for all your comments.

Reviewer 2 Report

Comments and Suggestions for Authors

I was very excited to be invited to evaluate the review article "Homonymous Hemi-Atrophy of Macular Ganglion Cell Layer as a Marker of Retrograde Neurodegeneration in Multiple Sclerosis – A Literature Review" by Larisa Cujbă and colleagues. This is a highly valuable topic, and the authors have done an excellent job summarizing the existing literature.

The manuscript provides a thorough review of cases and studies utilizing OCT to assess evidence of multiple sclerosis, and it effectively covers a wide range of relevant research. However, despite the comprehensive nature of the review, it is noticeably lacking in visual examples of OCT imaging related to the various types of MS presentations discussed. Given that this paper focuses on an imaging method, the absence of illustrative examples is a significant, critical omission.

To enhance the quality and utility of the review, it is essential to include case examples with OCT images for each pattern described. Visual representations will greatly aid in the understanding and application of the information presented.

Therefore, I recommend that the manuscript be revised to incorporate these crucial visual elements. I believe that with the addition of relevant OCT images, this review could make a significant contribution to the field.

Thank you for the opportunity to review this important work.

Comments on the Quality of English Language

Minor improvements are need to improve the clarity of exposition. I am certain if the authors chose to revise their manuscript copy editing will further improve the work. 

Author Response

I was very excited to be invited to evaluate the review article "Homonymous Hemi-Atrophy of Macular Ganglion Cell Layer as a Marker of Retrograde Neurodegeneration in Multiple Sclerosis – A Literature Review" by Larisa Cujbă and colleagues. This is a highly valuable topic, and the authors have done an excellent job summarizing the existing literature.

The manuscript provides a thorough review of cases and studies utilizing OCT to assess evidence of multiple sclerosis, and it effectively covers a wide range of relevant research. However, despite the comprehensive nature of the review, it is noticeably lacking in visual examples of OCT imaging related to the various types of MS presentations discussed. Given that this paper focuses on an imaging method, the absence of illustrative examples is a significant, critical omission.

To enhance the quality and utility of the review, it is essential to include case examples with OCT images for each pattern described. Visual representations will greatly aid in the understanding and application of the information presented.

Therefore, I recommend that the manuscript be revised to incorporate these crucial visual elements. I believe that with the addition of relevant OCT images, this review could make a significant contribution to the field.

We thank you for your valuable suggestions. We have added OCT images with case examples.

Thank you for the opportunity to review this important work.

Comments on the Quality of English Language

Minor improvements are need to improve the clarity of exposition. I am certain if the authors chose to revise their manuscript copy editing will further improve the work. 

We kindly thank you for your comments.

Round 2

Reviewer 1 Report

Comments and Suggestions for Authors

The authors of the article made all the necessary corrections according to the comments in the first round of review, so I, for my part, can recommend this manuscript for publication.